# Innovative Nanostructured Fillers for Dental Resins: Nanoporous Alumina and Titania Nanotubes

**DOI:** 10.3390/biomedicines11071926

**Published:** 2023-07-07

**Authors:** Roberto Eggenhöffner, Paola Ghisellini, Cristina Rando, Eugenia Pechkova, Tercio Terencio, Barbara Mazzolai, Luca Giacomelli, Katia Barbaro, Stefano Benedicenti

**Affiliations:** 1Department of Surgical Sciences and Integrated Diagnostics (DISC), Genova University, Corso Europa 30, 16132 Genova, Italy; paola.ghisellini@unige.it (P.G.); lu.giacomelli6@gmail.com (L.G.); benedicenti@unige.it (S.B.); 2Biostructures and Biosystems National Institute, Viale delle Medaglie D’Oro 305, 00136 Rome, Italy; eugenia.pechkova@gmail.com (E.P.); katia.barbaro@izslt.it (K.B.); 3Laboratories of Biophysics and Nanotechnology, Department of Experimental Medicine (DIMES), Genova University, Via A. Pastore 3, 16132 Genova, Italy; 4Istituto Italiano di Tecnologia (IIT), Viale Rinaldo Piaggio 34, 56025 Pontedera, Italy; tercio.terencio@conssp.it (T.T.);; 5Istituto Zooprofilattico Sperimentale Lazio e Toscana “M. Aleandri”, 00178 Rome, Italy

**Keywords:** dental composite materials, TiO_2_ nanotubes, anodized porous alumina, fillers

## Abstract

The possibility of improving dental restorative materials is investigated through the addition of two different types of fillers to a polymeric resin. These fillers, consisting of porous alumina and TiO_2_ nanotubes, are compared based on their common physicochemical properties on the nanometric scale. The aim was to characterize and compare the surface morphological properties of composite resins with different types of fillers using analytical techniques. Moreover, ways to optimize the mechanical, surface, and aesthetic properties of reinforced polymer composites are discussed for applications in dental treatments. Filler-reinforced polymer composites are the most widely used materials in curing dental pathologies, although it remains necessary to optimize properties such as mechanical resistance, surface characteristics, and biocompatibility. Anodized porous alumina nanoparticles prepared by electrochemical anodization offer a route to improve mechanical properties and biocompatibility as well as to allow for the controlled release of bioactive molecules that can promote tissue integration and regeneration. The inclusion of TiO_2_ nanotubes prepared by hydrothermal treatment in the resin matrix promotes the improvement of mechanical and physical properties such as strength, stiffness, and hardness, as well as aesthetic properties such as color stability and translucency. The surface morphological properties of composite resins with anodized porous alumina and TiO_2_ nanotube fillers were characterized by Atomic Force Microscopy (AFM), Scanning Electron Microscopy (SEM), and X-ray chemical analysis. In addition, the stress–strain behavior of the two composite resins is examined in comparison with enamel and dentin.

## 1. Introduction

Resin-based dental composites have markedly evolved since they were first introduced in dentistry more than 50 years ago [1] and remain among the most widely adopted, with hundreds of millions of composite treatments being placed every year worldwide [2]. Reinforcing fillers have been involved in some of the most important progress in dental resins achieved during the last decade, aiming to reduce the filler size to produce materials with improved mechanical properties and biocompatibility that can be easily and effectively polished. As a result, they are expected to be more resistant to wear and mechanical damage [3]. Inorganic filler-based composite resins are a type of dental restoration material that is used to fill cavities or small defects in the teeth. Fillers for dental restorations may be made from a variety of materials other than composite resins, including metals and ceramics. Metallic fillings such as amalgams have been in use for many years and are known to be durable, despite concerns about toxicity. Ceramic fillings are made from porcelain and can be matched to the surrounding teeth. Composite resins are becoming an increasingly popular choice of fillings because they can be color-matched, are less noticeable than metal fillings, and show more flexible adaptations. Improvements in durability and mechanical response remain a problem for the latter alternative. Aluminum and titanium are among the most popular elements for dental and orthopedic implants. Recently, it has been suggested that surface anodization may overcome degenerative conditions by the interposition of a thin buffer layer. However, the detailed effects of this oxide layer with respect to its thickness and roughness are still under discussion. Anodized aluminum has been used in medical applications as well, mainly to exploit its ability to resist the corrosion caused by highly corrosive body fluids. Recent progress in nanostructures has led to the implementation of anodized nanoporous alumina (APA) and TiO_2_ nanotubes, which are two types of nanostructured materials showing great promise for applications in the medical and biomedical fields. APA and TiO_2_ nanotubular materials show a higher surface-to-volume ratio due to their unique nanoscale properties. Therefore, they are ideal for use in applications such as tissue engineering. Furthermore, nanostructured materials often show improved mechanical properties compared to their bulk counterparts. For instance, APA and TiO_2_ nanotubes have been shown to have higher strength and stiffness than their bulk counterparts. This property makes them ideal for use in applications where high strength and stiffness are required, such as bone implants. Specifically, the long-term stability of implant integration and influence of the presence of APA and TiO_2_ nanotubes on metal surfaces must be carefully characterized [4]. Many combinations of resins and fillers have been tested to date; overall, however, they have not shown conclusive advantages over “traditional” amalgams [5]. According to meta-analysis results, there is low-quality evidence suggesting that resin composites lead to higher failure rates and a higher risk of secondary caries than amalgam restorations [6]. However, other studies support the effectiveness of resin composites for dental restorations [7,8,9], their antimicrobial and adhesive effects [10,11], and their enhanced biocompatibility [12]. It is well known that the process of osseointegration, consisting of the migration and proliferation of osteoblastic cells and the subsequent synthesis, deposition, and mineralization of the bone matrix, is enormously influenced by the roughness and chemical composition of the surfaces in question, which play a key role in optimizing the biological response. Thus, the morphology of APA and TiO_2_ nanotubes has been investigated through Atomic Force Microscopy (AFM), which is considered the gold standard for evaluation of 3D morphology of material surfaces both on the micro-scale and down to nano-sized resolutions [13]. The consequence of better adhesion lies in stronger and more stable contacts between the cells; initially, the less rough surfaces show greater proliferation than the more irregular ones, while in the long run the situation shifts in favor of more uneven surfaces. The optimal level of roughness has not yet been defined. In the literature, a value of 1.5 μm is considered acceptable when compared with other higher/lower values in the case of interfaces between an anodized metal surface and soft tissues, while a much lower value of around 200 nm is regarded as a good roughness limit with hard tissues. Low surface roughness gives rise to unsatisfactory results, particularly in the long term (30 days), making it critical for most treatments. Surface roughness and the morphological properties of restorative polymeric resins charged with inorganic fillers are discussed in terms of the porosity of alumina and the interlinking offered by titania nanotubes as fillers in composites. In particular, in the oral environment the surface roughness of restorative composites favors the adsorption of salivary proteins, adhesion of bacteria, and formation of bacterial plaque, which can cause infiltration and secondary caries, as a large surface with high roughness and waviness can hardly be cleaned in an optimal manner during the normal dental hygiene operation of tooth-brushing [14].

## 2. Materials and Methods

### 2.1. APA Preparation

APA layer preparation employed high-purity (>99%) Al foils (Goodfellow, Goodfellow Cambridge Limited, Huntingdon, UK) with a thickness of 250 µm, which were used as starting materials for the electrolytic synthesis of APA. The metallic surface was degreased in ethanol with an analytical purity grade of 99%. Afterward, the foils were kept at 400 °C for 4 h in an oven for thermal annealing. After this primary procedure of partial oxidation, the aluminum was cooled to room temperature for electrochemical treatment. This procedure was performed in a temperature-controlled apparatus, with a freezing mixture of ice and salts used to control the temperature and prevent the metal from burning. The apparatus used in this process was an electrochemical cell made up of two electrodes placed vertically at approximately 1.5 cm, using a 0.4 M aqueous phosphoric acid electrolyte at 110 V and a current density of 15 A/dm^2^. After electrochemistry, the samples were cleaned and maintained at 400 °C for 4 h in an oven for thermal annealing. The pore size resulted in pores of uniform size over a long-range scale, i.e., several micrometers, even in preparations based on only a single-step anodization process. The resulting pattern of pores, however, showed deviations from the regular hexagonal shape. A mean pore diameter in the range of 50–100 nm and interpore distances of 150–200 nm were obtained in the present case. The oxide coating was set free by dissolving the aluminum substrate in saturated CuCl_2_, the pore bottoms were opened in phosphoric acid, and the APA membranes were pulverized in a planetary ball mill (PM 100, Retsch, Nordrhein-Westfalen, Germany) to obtain APA powder with particles of 5–10 μm diameter, as confirmed by scanning electron microscope inspection.

### 2.2. TiO_2_ Nanotube Preparation

The TiO_2_ nanotubes were synthesized using the hydrothermal procedure, currently the most widely adopted, in order to constitute a standard preparation [15]. Commercial nanopowders of TiO_2_ (anatase phase) with an average particle size below 25 nm (TEM) and trace metals at ≥99.7% were provided by Aldrich-Merck KGaA, Darmstadt, Germany. Nanoparticles (500 mg) were dispersed in 50 mL of a 10M sodium hydroxide solution, then the suspension was stirred for 2 h at room temperature and subsequently was transferred to an oven for hydrothermal treatment at 160 °C for 24 h. The obtained product was treated in a 0.1 M HCl solution for 3 h, then the suspension was centrifuged and the solid product was washed with distilled water to lower the pH to 6.7. The sample was dried at 80 °C for 24 h in the oven before final annealing at 400 °C for 2 h.

### 2.3. Preparation of Resin Composite Samples

The introduction of Bisphenol A, i.e., the well-known glycidyl glycerolate dimethacrylate (BisGMA) [16], has contributed to the growing success of dental resin composites, offering the opportunity to replace the metal amalgams. Indeed, amalgams continue to be used in spite of the general consensus regarding the toxicity of mercury. In addition, there may be critical safety issues with regard to Methacrylate-based dental resins. The latter composites are widely used, and in particular are considered a test-bench for testing the qualities offered by new fillers to match requirements for odontoiatric applications. The composite resin with APA filler was prepared by mixing 50 mg of Bis-GMA as the main organic matrix monomer with 2-hydroxyethyl methacrylate as a diluent to reduce shrinkage, along with camphorquinone as a photoinitiator [17] and 50 wt% of APA powder after its separation from the supporting aluminum substrate. HEMA sonication were used to assist in expediting the spatulation of the mixture, which was placed on a glass slide and immediately irradiated with a lamp (3M ESPE Elipar Free Light 230 V/50/60 Hz) for 30 s to obtain complete photopolymerization. No silane bonding agent was used, allowing the effects of the new fillers to be investigated without interference from other effects. The same procedure was repeated with the TiO_2_ nanotube powder prepared using the hydrothermal treatment discussed in the previous section, with the same percentage by weight of filler with respect to bias-GMA. Furthermore, a quantity of 50 mg of Tetric EVO Flow (TEF) by Ivoclar was used to prepare representative samples of a conventional resin for comparison. The procedures adopted for preparation of the sample composite resins with both APA and TiO_2_ fillers resembled the technique usually adopted by dentists in the clinical application of such materials in the oral cavity. Our aim was to characterize the surface morphological properties of the two composite resins with APA and TiO_2_ nanotubes as fillers and to compare their morphological and mechanical proprieties.

### 2.4. Atomic Force Microscopy (AFM), Scanning Electron Microscopy (SEM) Characterization, and X-ray Chemical Analysis

SEM with microanalysis was performed using a Carl Zeiss AG/Oxford instruments EVO MA10. This instrument works on a sample by scanning it with a focused beam of electrons. It uses Silicon Drift Detector microanalysis, allowing the constitutive materials that make up the sample to be identified. The instrument works within a 0.2–30 kV acceleration voltage range, and is provided with a secondary electron detector and a backscattered electron detector. Samples were analyzed through a Dual Beam–FIB/SEM FEI HELIOS NANOLAB 600i by combining a 350 V–30 kV SEM and a 500 V–30 kV FIB, as it is the preferred solution for 3D microscopy and analysis for material characterization, industrial failure analysis, and process control applications. This instrument additionally enables low energy FIB operation, FE SEM analysis, STEM microscopy, EDS, WDS, and EBSP analysis, and is provided with OmniprobeTM sample extraction for microanalysis. The Innova Atomic Force Microscope (AFM by Bruker) allows high-resolution imaging and 3D topographic maps of a material surface to be obtained, and can quantify surface-related parameters such as roughness. The images were acquired in tapping mode with a cantilever (AFM Probe) “umasch HQ:NSC18/Cr-Au” using a force constant of 2.8 N/m (1.2–5.5 N/m), resonant frequency of 75 kHz (60–90 kHz), length of 225 µm (1–230 µm), width of 27.5 µm (24.5–30.5 µm), and thickness of 3 µm (2.5–3.5 µm) [18].

### 2.5. Mechanical Measurements

The stress–strain behavior of dental resin fillers is an important consideration in their design and use. Dental resin fillers are materials used in odontoiatry to restore damaged or decayed teeth. When the resin is cured, it forms a hard and durable surface that can withstand the forces of biting and chewing. Thus, in dental applications, the main stress and strain measurements are typically performed under compression, and less frequently under tension or bending. The stress–strain behavior of dental resin fillers depends on a variety of factors, including the composition of the resin and filler, the curing, and the loading conditions. Generally, dental resin fillers exhibit linear elastic behavior up to a certain load limit, after which they begin to exhibit plastic deformation and ultimately fracture. To optimize the performance of dental resin fillers, it is important to understand their stress–strain behavior and design them to withstand the forces of biting and chewing. This can be accomplished through careful selection of resin and filler materials as well as by controlling the curing conditions to achieve the desired mechanical properties. In our case, we checked the behavior under compression of TiO_2_ nanotube resin composites. An Instron 4464 was used for compression stress–strain measurements. This instrument is suitable for static mechanical tests in compression or tensile mode within a single frame. A cell load with a limit of ±1 kN was employed, and the stress was applied by compressing disks with dimensions of 5 mm in diameter and 2 mm in height.

## 3. Results and Discussion

The process of aluminum anodization has long been used to protect metal against corrosion, as well as for decorative purposes. As described above, the formation of APA films is obtained from the electrochemical oxidation (anodization) of aluminum in acid electrolytes in the conditions that balance the growth at the metal–film interface and the dissolution at the pore–electrolyte interface. Thus, the surface films are formed with thickness limited to 1–30 µm depending upon the acid selected in anodization. In this case, the anodization voltage (110 V) was selected to produce larger pores with diameters of 50–120 µm and interpore distances of 100–200 µm, as shown in Figure 1a. Pore sized is a major issue in achieving efficient interaction with the organic component in the polymerization [19]. Thickness values from 3.5 to 5 µm can be observed in Figure 1b. The morphology of the SEM images in Figure 1a,b are closely aligned with the results from the literature [20,21,22] for APA materials prepared with similar settings.

TiO_2_ nanotubes are one-dimensional nanostructures that have a high surface area, tunable band gap, and enhanced photocatalytic activity. Three main routes can be adopted for preparing TiO_2_ nanotubes. In the template-assisted method, TiO_2_ nanotubes are prepared by employing various templates to control the shape and size of the nanotubes themselves. Examples of template-assisted methods for TiO_2_ nanotubes include multi-walled carbon nanotubes (CNT), alumina template membranes and electrospun water-soluble nanofibers. A carbon nanotube template is a good choice for improving the photocatalytic performances. The alumina template membrane can be used to fabricate TiO_2_ nanotubes via hydrolysis with water vapor, which can produce smaller and thinner nanotubes than sol–gel template synthesis. Electrospun water-soluble polyvinyl alcohol (PVA) nanofibers can be used to prepare thoroughly mesoporous TiO_2_ nanotubes with intact morphology that can enhance the photocatalytic activity under visible light.

In addition to the above template routes, electrochemical anodic treatment of the Ti surface is another preparation route; as described in the introduction, it is convenient for implant applications, as it produces an ordered alignment of nanostructured TiO_2_ nanotubes with a high aspect ratio. Finally, the hydrothermal treatment constitutes the easiest route for obtaining a nanotube morphology with random alignment in powder form which is able to readily mix with the other organic components. Further, a number of modifications can be easily used to enhance the attributes of titanium nanotubes by changing parameters such as the annealing temperature, annealing time, and molar concentration of sodium hydroxide solution, among others. The achievement of TiO_2_ nanotubes with the appropriate morphology is shown in Figure 2, in which the transformation from the initial anatase TiO_2_ powder (Figure 2a) to the needle shape of the TiO_2_ nanotubes (Figure 2b) is reported. Scanning electron microscopy coupled with an energy dispersed X-ray microprobe was used for SEM/EDX examination of the inorganic filler extracted from the composite.

Surface morphology plays a crucial role in assessing the properties of practical nanofillers in composite resins made with an inorganic nanostructured component in a polymeric environment [23]. Thus, we studied the surfaces of the alumina and titania fillers in the same polymeric material discussed above. The surface morphology of the composite resins with the two fillers selected for mechanical improvement was characterized using SEM and AFM. In Figure 3, the AFM morphological analysis of the two resin samples based on APA and titania nanotubes is reported (Figure 3a and Figure 3b, respectively).

The composite surface with alumina filler is characterized by an irregular distribution of large grains and protrusions. The TiO_2_ resin surface shows a more regular distribution of protrusions and valleys extending in length over the whole explored surface (10 µm^2^). This behavior is confirmed by the profiles reported in Figure 4 for the APA resin and Figure 5 for the TiO_2_ nanotube resin. The detected profiles are shown by the blue colored lines in Figure 3.

The depth extending from protrusions to valleys in Figure 5 (TiO_2_ nanotubes) is about half that in Figure 4 (APA). Further, the lower profile in Figure 5 is detected along the path on the top of the protrusions, in which the profile exhibits only small corrugations.

To further confirm these morphological differences, the larger corrugations in the APA composite resins obtained as SEM micrographs are shown in Figure 6. The root mean square RMS of the roughness along the line transverse to the protrusions and valleys is equal to 620 ± 50 nm, and is equal to 300 ± 40 nm along the direction of the crests (the solid and broken lines at left and right scales of Figure 4, respectively). The average roughness value for the resin with titanium nanotubes along the line transverse to the protrusions and valleys is equal to 180 ± 30 nm, and to 140 ± 20 nm along the direction of the protrusions (the solid line and circle curves at the left and right scales of Figure 5, respectively). Values up to 750 nm were obtained (averaged values for areas compared to that considered in this work) for the conventional resins, with an empirical limit to due to potential bacterial colonization attacks that may occur if the RMS exceeds values in the range of 200–300 nm.

These findings indicate potential drawbacks of using APA dental fillers; their high surface area and porosity can make them more prone to adsorbing biological fluids or water, which can weaken the material and reduce its mechanical properties. In addition, the nanoporous structure of alumina particles can make them more difficult to process and incorporate into dental composite materials while reducing their high surface porosity. These requirements may increase the complexity and cost of manufacturing such material. While APA composites cannot be regarded as universal restorative materials in their current formulation, they could benefit from possible modifications while retaining the advantages in terms of nanopore structure, which for instance could be exploited for drug release. The present formulation might alternatively include smaller fillers than APA in order to fill the gaps up to approximately 5 µm. Alternatively, a composite of hybrid heterogeneous filler formulation with nanosized (non-porous) fillers intermixed with nanoporous APA microfillers might be explored.

TiO_2_ nanotubes have been shown to lower surface roughness in comparison to APA and conventional resins; they can be viewed as a good compromise that favors the adhesion of the material to the tooth structure without presenting too many surface defects. Thus, the resulting roughness can improve the bond strength between the restoration material and the tooth. On the other hand, nanoparticles and nanotubes tend to agglomerate in clumps, which is detrimental to both mechanical and aesthetic performance because they become resistant to dispersion by organic solvents in the polymerization process [24]. Surface modification of nanoparticles as TiO_2_ nanotubes has been suggested to prevent agglomeration and improve compatibility with resin matrices, reducing the above range of surface roughness values to lower levels. One such route is represented by the silanization process, in which an ethanol–water solution of silanes is added directly to the filler mixture in order to decrease the flow values of fillers as TiO_2_ nanotubes within composite materials and improve their dispersion and bonding to the resin matrix. [25]. Previous studies have found a strong correlation between surface roughness and the mechanical properties of resin composite materials [26,27,28]. The surface roughness values detected for the APA and TiO_2_ nanotube fillers lead to surface indentations that can be treated by surface finishing. Therefore, smooth and aesthetically valid surfaces of these composites can be obtained by polishing. Alternative solutions have been reported to provide smoother surfaces as well, leading to decreased surface irregularities [29].

In Figure 7, an SEM micrograph of a representative area of the filler composite shows a cross-linked scaffold induced by nanotubes of the shape reported in Figure 3b with respect to the more regularly distributed aluminum content (see Figure 6). The AFM and EDS_SEM results are consistent with the high surface area and specific surface chemistry of the TiO_2_ nanotubes, which promote the formation of strong bonds between adjacent nanotubes. The reported pattern, very differently from the clumping agglomeration, provides a scaffold at the nanometric scale for mechanical reinforcement of the resin. Obviously, a compromise between the flowability offered by silanization and the formation of an interlinking pattern would be required for optimal composites.

In Figure 8, the stress–strain behaviour of the two resins with APA and TiO_2_ nanotubes as fillers is reported for comparison with the base resin without fillers and data from the literature on enamel and dentin [30]. In the latter reference, the mechanical properties of enamel and dentin were measured in compression tests using healthy human teeth specimens with the same dimensions, as in the present work for the composite resins. The curves in Figure 8 are always concave and upwards, meaning that the slope rises continuously without satisfying Hooke’s law. In no range in our experimental data do the curves behave in the linearly fashion shown by trend lines reported in Figure 8. The stress–strain curves show no decrease in elasticity, implying that fracture eventually occurs. The behaviour of the two fillers (the green and blue curves in Figure 8) is different in the low and high stress ranges, with a critical value of 110 MPa. As with TiO_2_ and APA, enamel proves stiffer than dentin in the range of stresses below 40 MPa and less rigid in the higher stress ranges.

The inset of Figure 8 shows that the curve for the TiO_2_ nanotubes is closer to the behaviour of enamel, whereas the APA and base resin are closer to the trend of dentin. Assessment of these results reveals that only the combined presence of both fillers can match the required mechanical properties of an optimal composite resin. The results further indicate that the implementation of the novel filler based on the interconnection provided by nanotubes in currently used dental materials represents a promising feature for the performance of high-strength materials in the stress–strain range shown. Namely, the highest bite compression stress in mastication is 30 MPa, apparent at the top of the scale in the inset of Figure 8 [31].

To date, these findings require further investigation and improvement [32] considering that, in particular, APA appears to be applicable as a drug provider. TiO_2_ nanotube implementations in more complex mixtures of filler composition are definitely more feasible based on our morphological results. For example, nanofibrillary filler materials could improve packaging density by connecting TiO_2_ nanotubes dental composite resins. TiO_2_ nanotubes could offer significant improvements when used in conjunction with small particles such as boron nitride nanosheets in resins, a route that might improve flowability and further enhance the mechanical and antimicrobial properties of dental resins [33]. A final requirement concerns the practical ease of handling the materials by dentists in order to provide the best results for patients in terms of the absence of inconveniences and long-term durability of the applications. Further research appears crucial in order to confirm the biocompatibility of nanomaterials such as APA and TiO_2_ nanotubes [34,35] used in composite resin fillings.

APA and TiO_2_ nanotubes share certain characteristics when used as fillers in dental resins. Both materials are non-toxic, making them good candidates for use in dental applications. Additionally, they have large surface areas and form strong chemical bonds with the resin matrix, improving the mechanical properties of the composite material. When used as fillers in dental resins, both APA and TiO_2_ nanotubes provide satisfactory hardness, strength, and wear resistance of the resulting composite material.

However, there are a number of differences between the two materials when used as fillers in dental resins. APA is usually used as a microfiller, as it has particle sizes around the micron range. On the other hand, TiO_2_ nanotubes are used as nanofillers, as their size is within the nanometer range. As a result, TiO_2_ nanotubes provide the correct reinforcement and efficient stress transfer to the resin matrix in the low stress range in which mastication action occurs. Results from the literature show a wide consensus about the need to combine nano- and micrometric fillers in order to meet the required mechanical strength. To this end, APA could be adopted in combination with TiO_2_ nanotubes, a suggestion for a possible future investigation, without excluding other fillers in an optimal mixture to meet aesthetic requirements [36].

## 4. Conclusions

APA and TiO_2_ nanotubes are payloads in dental resins that can support improved mechanical properties and other desirable features. While they share common properties such as biocompatibility and the ability to form strong chemical bonds with the resin matrix, there are differences in terms of their performance and suitability for different types of dental applications.

Globally speaking, the results of the present investigation support the conclusion that TiO_2_ nanotubes represent a paramount candidate for use in the composition of inorganic compounds, constituting an optimal nanometric filler for dental composite materials.

The present work confirms that the most appropriate recipe to match the large number of requirements of a final product in the market must necessarily be very complex and involve multiple compounds in order to fulfill contemporarily mechanical wear resistance, durability, biocompatibility, and aesthetic requirements for the performance of dental restorative materials.

## Figures and Tables

**Figure 1 biomedicines-11-01926-f001:**
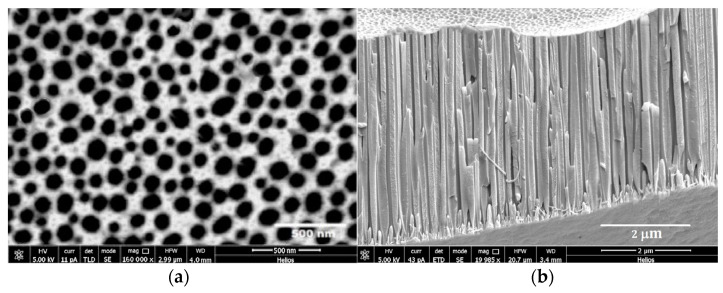
(**a**,**b**). High-resolution SEM images acquired at high beam energy (5 kV), showing (**a**) the APA surface and (**b**) a cross-section of the APA pores, after electrochemical treatment of the polished Al foil and before surface scraping to prepare the powder used as filler. The cross-sectional view shows the thin barrier layer next to the metal and beneath the outer layer of APA.

**Figure 2 biomedicines-11-01926-f002:**
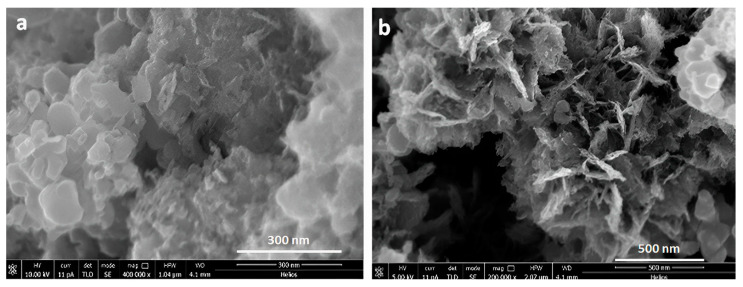
High-resolution SEM micrographs of TiO_2_ nanotubes acquired at high beam energy (10 keV): (**a**) Anatase TiO_2_ powder used for the hydrothermal treatment (1.5 × 1.0 µm^2^) and (**b**) the TiO_2_ after nanotube preparation (2.0 × 1.5 µm^2^), demonstrating the effectiveness of the hydrothermal method.

**Figure 3 biomedicines-11-01926-f003:**
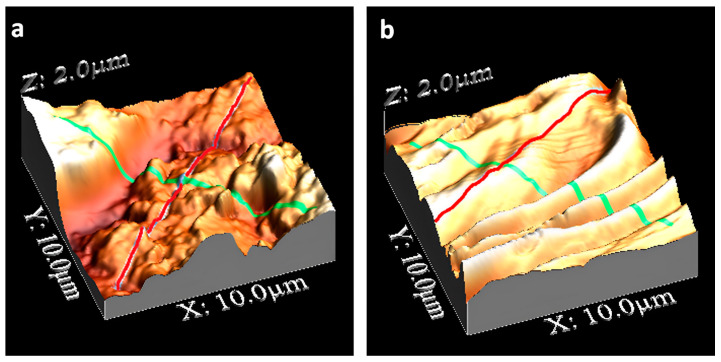
AFM imaging of composite resins of APA Al_2_O_3_ (**a**) and TiO_2_ nanotubes (**b**). Surface areas of 10 × 10 µm^2^ are displayed, the green lines cross the protrusions-valleys, the red lines are drawn on the ridge of a protrusion.

**Figure 4 biomedicines-11-01926-f004:**
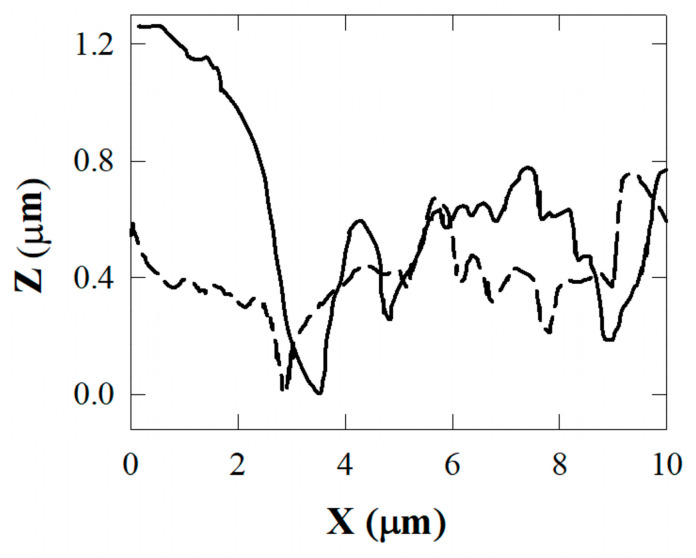
Profiles along the two lines in Figure 3a. The depth of the protrusions/valley observed in the APA composite is shown: the solid curve corresponds to the green lines reported in Figure 3a, the broken curve corresponds to the red line in the same figure.

**Figure 5 biomedicines-11-01926-f005:**
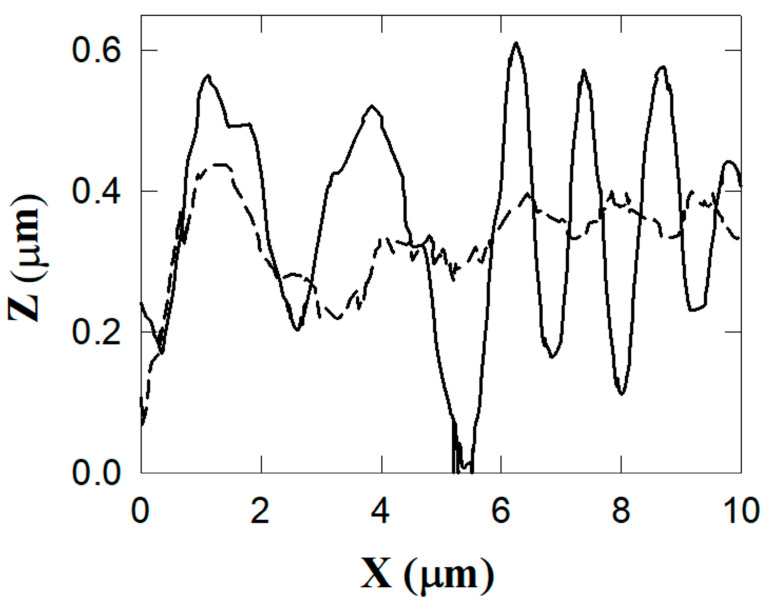
Profiles along the two lines in Figure 3b. The depth of the protrusions/valley observed in the titania composite is shown: the solid curve corresponds to the green lines reported in Figure 3b, the broken curve corresponds to the red line in the same figure.

**Figure 6 biomedicines-11-01926-f006:**
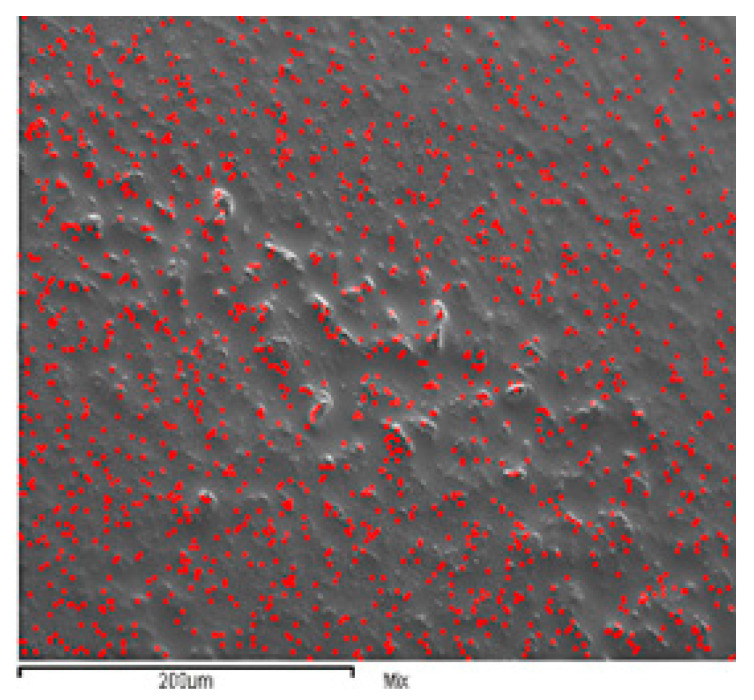
Surface of the resin composite with dispersed APA. The surface roughness is visible at the micrometric scale. The red points indicate the EDS signals from Al atoms.

**Figure 7 biomedicines-11-01926-f007:**
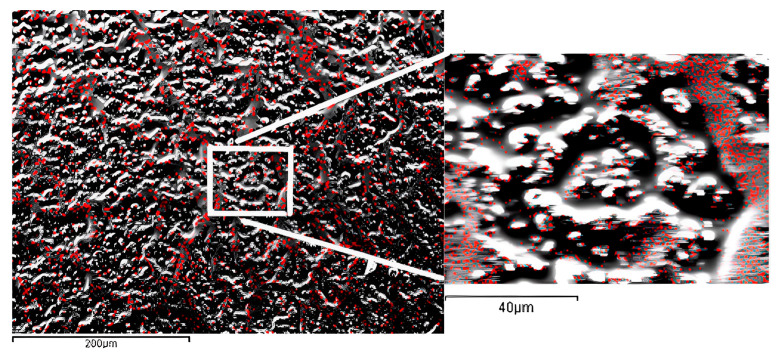
EDS-SEM micrograph of the surface of the resin composite with dispersed TiO_2_ nanotubes as filler. The red points indicate the presence of Ti signals. The surface appears structured, with the Ti distribution concentrated along lines visible at the micrometric scale reported, as shown in the detail of the white boxed area.

**Figure 8 biomedicines-11-01926-f008:**
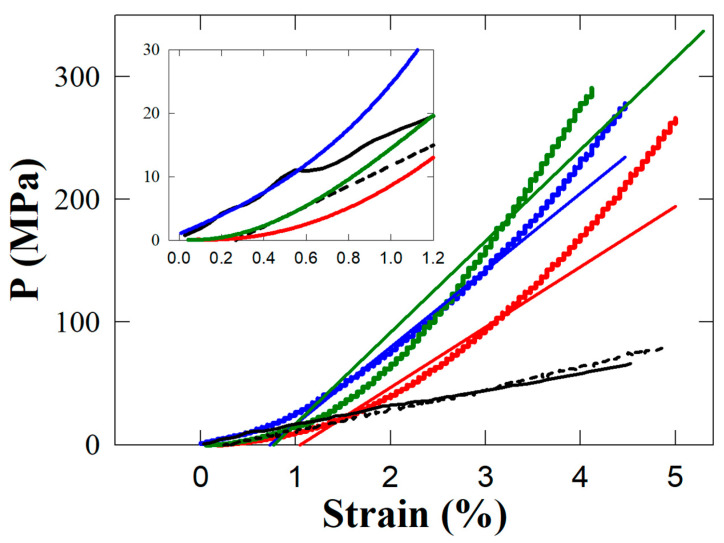
Stress–strain behaviour of the composite samples under compression. Red curve: base resin; blue curve: TiO_2_ nanotubes in resin; green curve: APA in resin. The curves for enamel (solid black) and dentin (dashed black) are reported for comparison purposes. The inset of the figure emphasizes the stress at very low deformations, and uses the same units as in the main figure. The red, green and blue lines are the trend straight lines showing the deviations from the linear elastic behaviour.

## Data Availability

Details on the calculations and graph presentations can be obtained from the corresponding author.

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
