# Peer review of "Innovative Nanostructured Fillers for Dental Resins: Nanoporous Alumina and Titania Nanotubes"

_biomedicines, 2023, doi:10.3390/biomedicines11071926_

Round 1
Reviewer 1 Report
The research manuscript entitled “Innovative nanostructured fillers for dental resins" by Eggenhoffner et al. for Biomedicine. This research work summarizes the physical characteristics of nanocomposite of resin with APA and TiO2, and also some of the information/data are very relevant however, some data are not well organized. I will recommend for major revision before acceptance for publication. Following concerns should be addressed:
1. Please edit the "Abstract" and define the objective of the research paper in the "Abstract". It is not clear what authors have investigated in this research. What
2. I the "Introduction" section authors have provided details information about anodized nanoporous alumina (APA), and TiO2, however it lacks information of other nanomaterials with resin. Therefore, please edit the "Introduction" with latest research articles where polymer resin with nanocomposites have been developed for dental applications. Following research paper can be cited: Dental Materials 2018, 34, e63-e72, Scientific Reports 2019, 9, 4921, and Nanomaterials 2020, 10(9), 1750, and other.
3. Page 4, line 173, please provide the closure of bracket " The Innovation ……(AFM by Bruker………as roughness".
4. Please provide proper caption for all figures. Figure 1, please provide (a) and (b). The scale bar is mot visible, please make it visible.
5. Please provide the SEM images of APA with same magnification of TiO2.
6. Please provide (a) and (b), as well as scale bar in Figure 2.
7. Please re-plot the Figure 3, and 4 with different color, which will help to easily identify the profile. Please provide the proper caption for both the figures (i.e., 3, and 4).
8. In figure 5, please provide the complete information in the caption, Please explain the meaning of "Red" dots.
9. Please provide the quantitative value for mechanical properties of the resin-APA, and resin-TiO2 composites.
Please check the english.
Author Response
Please consider the file submitted.

Reviewer 2 Report
Dear authors,
I have several suggestions to improve your manuscript:
1. the title of the manuscript should be modified, in the present form it is not precise enough and seems like the title of a systematic review nor original article
2. the abstract should be re-written; introduction part is too long, there is no number of the sample, results and conclusion are not clear
3. there is no aim of the study on the end of introduction part, please add
4. please specify the power analysis and the sample size calculation process
5. please specify the final number of the samples
6. if I got correctly, you just have presented qualitative results, but quantitative results should be presented
7. it would be much clearer if the results and the discussion were written separately
8. conclusion should be re-written, should be clear and concise
Author Response
Please consider the file submitted

Reviewer 3 Report
The manuscript entitled ,,Innovative Nanostructured Fillers for Dental Resins” is focused on the production and characterization of nanostructured fillers for dental composites. Two nanostructured fillers were synthesized and characterized: anodized nano-porous alumina (APA) and TiO2 nanotubes. These nanostructured fillers were used to prepare BisGMA based composites which were also characterized. Overall, it is an actual research of great interest for the specialist in the field. Experimental setup is well organized and conducted. Experimental results are mostly good but unfortunately they are presented in a poor manner and require improvements. Conclusions look like an extra-discussion with references to literature. Therefore, the manuscript needs significant corrections, completions and improvements. Please revise the manuscript according to the comments below:
Comment 1) SEM characterization of APA microstructure is missing. SEM images of APA must be introduced and discussed in text. You mention SEM investigation of APA powder in lines 118 -119. Therefore, APA microstructure discussion paragraph including SEM images must be introduced at Results and Discussion before TiO2 nanotubes paragraph.
Comment 2) Figure 1 caption is unclear and must be revised. I understand from the text that Figure 1a is the SEM image for starting TiO2 powder and Figure 1b is the SEM image of the TiO2 nanotubes resulted after hydrothermal treatment.
Comment 3) Lines 172 – 174: AFM method must be completed with specific details such as: Used cantilever type, producer, resonant frequency and force constant; AFM working regime tapping or contact mode?; scanning rate.
Comment 4) AFM images in Figure 2 are presented in a very poor manner that reduces considerably their quality, x, y, z axis of presented 3D topographic images are completely invisible. Therefore your allegation that scanned area is 10 μm x 10 μm is not sustained by the presented images. In consequence Figure 2 must be revised and improved. I recommend to present 2D topographic image for a better observation of the morphologic details beside its 3D representation which better reveals the topographic aspects.
Profiles in Figure 3 are ok. Profiles in Figure 4 are difficult to be followed because of two Z axis unspecified which of the profiles belongs.
Comment 5) Figure 5 must be completed with SEM electron image corresponding to the presented EDS map. I guess that red spots in the EDS map belongs to Al, fact must be mentioned in the figure caption. Also it is recommended to present an EDS spectrum and complete elemental composition of the composite.
Comment 6) Figure 6 must be completed with SEM electron image corresponding to the presented EDS map. Also it is recommended to present an EDS spectrum and complete elemental composition of the composite.
Comment 7) Conclusion is not proper and requires a major reorganization. Lines 301 – 319 must be moved at discussion section. The conclusions must be presented shortly and concise in several paragraphs pointing out the findings within research.
Comment 8) Mechanical properties testing results (Compressive Strength) completely missing from the text. Testing curves must be presented and the obtained values properly discussed and compared to the data in literature.
Comment 9) I recommend to be careful when discuss about antimicrobial properties because you do not effectuate any test in that way.
Minor editing of the English Language is recommended.
Author Response
Please consider the file submitted.

Round 2
Reviewer 1 Report
I will recommend for publication now.
Author Response
We thank the reviewer for the positive comments.
Reviewer 3 Report
The requested major corrections and completions were well effectuated.
The requested information regarding elemental maps in Figures 6 and 7 in current version of the manuscript were properly presented.
This comment refers only to the cover letter and do not affects the manuscript: EDS data presented in cover letter are poor: both spectra looks truncated; data presented in the table are ambiguous and do not present any measuring unit. It is expected to present the elemental composition in atomic or weight percent. For example: you mention Au 652 for APA sample - why gold? (perhaps sample was gold metalized for a better visibility - in such case Au should be substracted from the EDS spectra to avoid sample composition perturbation); what 652? It is not necessary to respond at these questions, please be more careful for the future. It is good that you do not present these EDS spectra in the manuscript.
Some minor corrections are mandatory:
1) Please take care to the figures re-numbering, there are large text parts where the figures numbers are not updated after first revision.
2) Lines 423 - 426 must be moved from Conclusions to the Discussion section. It is not good introducing references into the Conclusions section.
3) Use point as decimal separator instead of comma including in figures.
Author Response
We thank reviewer's comments. We have corrected as required 1) the Figure calls by re-numbering; 2) we have moved four lines from conclusion to discussion and changed comma to decimal separators in the figures.